# Genome-Wide Association Study Identified Candidate Genes for Alkalinity Tolerance in Rice

**DOI:** 10.3390/plants12112206

**Published:** 2023-06-03

**Authors:** Lovepreet Singh, Rajat Pruthi, Sandeep Chapagain, Prasanta K. Subudhi

**Affiliations:** School of Plant, Environmental, and Soil Sciences, Louisiana State University Agricultural Center, Baton Rouge, LA 70803, USA; lsingh@agcenter.lsu.edu (L.S.); rpruthi@agcenter.lsu.edu (R.P.); scgdt@missouri.edu (S.C.)

**Keywords:** abiotic stress, alkalinity tolerance, candidate genes, genome-wide association study, *Oryza sativa*, seedling stage, single nucleotide polymorphism

## Abstract

Alkalinity stress is a major hindrance to enhancing rice production globally due to its damaging effect on plants’ growth and development compared with salinity stress. However, understanding of the physiological and molecular mechanisms of alkalinity tolerance is limited. Therefore, a panel of *indica* and *japonica* rice genotypes was evaluated for alkalinity tolerance at the seedling stage in a genome-wide association study to identify tolerant genotypes and candidate genes. Principal component analysis revealed that traits such as alkalinity tolerance score, shoot dry weight, and shoot fresh weight had the highest contribution to variations in tolerance, while shoot Na^+^ concentration, shoot Na^+^:K^+^ ratio, and root-to-shoot ratio had moderate contributions. Phenotypic clustering and population structure analysis grouped the genotypes into five subgroups. Several salt-susceptible genotypes such as IR29, Cocodrie, and Cheniere placed in the highly tolerant cluster suggesting different underlying tolerance mechanisms for salinity and alkalinity tolerance. Twenty-nine significant SNPs associated with alkalinity tolerance were identified. In addition to three alkalinity tolerance QTLs, *qSNK4*, *qSNC9*, and *qSKC10*, which co-localized with the earlier reported QTLs, a novel QTL, *qSNC7*, was identified. Six candidate genes that were differentially expressed between tolerant and susceptible genotypes were selected: LOC_Os04g50090 (Helix-loop-helix DNA-binding protein), LOC_Os08g23440 (amino acid permease family protein), LOC_Os09g32972 (MYB protein), LOC_Os08g25480 (Cytochrome P450), LOC_Os08g25390 (Bifunctional homoserine dehydrogenase), and LOC_Os09g38340 (C2H2 zinc finger protein). The genomic and genetic resources such as tolerant genotypes and candidate genes would be valuable for investigating the alkalinity tolerance mechanisms and for marker-assisted pyramiding of the favorable alleles for improving alkalinity tolerance at the seedling stage in rice.

## 1. Introduction

Alkalinity stress drastically reduces rice yield. About 850 million hectares of land are affected by salinization–alkalization and 434 million ha of this land are exposed to alkaline stress [1]. Alkalization of land is increasing every year due to poor irrigation management and climate change [2]. Rice is a staple food for more than 3.5 billion people all over the globe [3]. Due to the exponential population growth in developing countries, rice production in abiotic-stress-affected areas needs to be enhanced for regional and global food security. However, progress in developing alkaline-tolerant varieties is very slow due to the genetic complexity of the alkalinity stress tolerance mechanisms. Therefore, research on rice’s alkalinity tolerance has great relevance for increasing global rice production.

Plants possess various physiological and molecular mechanisms for adaptation to an alkaline environment, and these mechanisms are controlled by the expression of specific stress-related genes [4,5]. Alkaline stress is more harmful to plants than saline stress, but there is a wide range of variations for alkalinity tolerance [6,7]. Alkaline stress is primarily caused by HCO_3_^−^ and CO_3_^2−^, as well as high pH [8]. These anions, alongside high pH, might be responsible for different plant adaptation strategies under alkaline stress. Alkali soils have pH values of 8.5–10.0, exchangeable sodium > 15, and electrical conductivity < 4000 micromhos per cm at 25 °C. Soil alkalinity restricts the growth of rice at all stages by decreasing nutrient availability, disrupting ionic balance, and increasing osmotic pressure, especially under high pH [9,10]. Plants need to cope with the high pH of soil or water under alkaline stress in addition to ionic and osmotic stresses [11]. High concentrations of Na^+^ in alkaline soils disrupt the homeostasis of minerals such as K^+^ and consequently, affect cellular metabolism by altering cytoplasmic strength [12]. Water and potassium uptake decrease due to Na^+^ accretion in the roots under excessive Na_2_CO_3_ around the rhizosphere. The precipitation of iron and phosphorus under a high pH environment causes deficiencies in these nutrients resulting in the wilting of plants [13,14]. Alkalinity-tolerant plants can sequestrate Na^+^ in vacuoles to enhance their tolerance to high concentrations of ions [15,16]. The Na^+^/H^+^ antiporter in the plasma membrane and HKT family of transporters help to prevent sodium from entering the shoots resulting in a lower sodium/potassium ratio [11,17]. 

High pH under alkaline stress negatively impacts the root system leading to reduced root surface area and impaired function [18]. To cope with these challenges, rice plants promote the accumulation of osmolytes and organic acids, which act as buffers and help maintain intracellular pH stability and ionic balance [19,20]. Ionic balance and nutrient uptake are associated with root structure as plants with vigorous root systems can maintain their growth under a stressful environment [21]. Alkalinity-stress-induced damage to root cells was reported to be influenced by the accumulation of reactive oxygen species and regulation of cell-death-related and cell-death-suppressor genes [22]. Similarly, plants with larger and deeper root systems can accumulate more Fe under alkaline stress [23]. The over-expression of genes such as *OsIRO2*, *OsIRT1*, *OsNAS1*, *OsNAS2*, *OsYSL15*, and *OsYSL2* also allows plants to efficiently take up iron and improve alkalinity tolerance [23].

Alkalinity tolerance in rice is a complex trait and multiple quantitative trait loci (QTL) control this attribute [24]. Most of the research related to alkaline stress tolerance in rice is in the primary stage of QTL mapping. Several major QTLs and identified candidate genes (LOC_Os03g59730; LOC_Os09g32860; LOC_Os10g35170) within these QTLs were identified using whole genome sequencing on chromosomes 3, 9, and 10 [2,7]. A major QTL, *qSNC3*, with two candidate genes (LOC_Os03g62500; LOC_Os03g62620), was detected [15]. A few other alkalinity-tolerance QTL studies were conducted at the seedling and germination stage of rice [24,25,26,27]. A few valuable genes associated with alkalinity tolerance have been identified. The role of LSD1-like zinc finger protein (*OsLOL5*), a nucleus thylakoid protein (*OsY3P1*), a potassium channel protein (*OsAKT1*), a snf2 family gene (*OsALT1*), a calcium-dependent protein kinase (*OsDM13*), and a gene encoding an inorganic phosphatase (*OsPPa6*) have been shown to improve alkalinity tolerance in rice [28,29,30,31,32,33]. Conducting a genome-wide association study (GWAS) is the widely used approach for providing insights into the molecular genetic basis of complex traits and identifying genes or molecular markers associated with target traits for crop improvement [34,35,36]. Compared to traditional QTL mapping, which involves examining the inheritance patterns of a single genetic trait, GWAS looks at the occurrence of variants in natural populations with high resolution [37]. Li et al. (2020) [38] identified a major QTL, *qAKT11*, for alkalinity tolerance in a GWAS study. A major QTL, *qSNC3*, and a candidate gene, *OsIRO3*, were detected for alkalinity tolerance [12]. Eight candidate genes conferring alkalinity tolerance at the germination stage were identified in another rice GWAS study [39].

In this study, we evaluated a diverse panel of 184 *indica* and *japonica* rice genotypes for alkalinity tolerance at the seedling stage to evaluate the genetic variation and population structure. To overcome the narrow genetic diversity, rice cultivars released in the United States during the 20th century and cultivars from IRRI were included in this study. The specific objectives of this study were to: (i) screen the *japonica* and *indica* rice genotypes for seedling-stage alkalinity tolerance, (ii) investigate the genetic variability, and population structure of the diversity panel, and (iii) identify QTLs and candidate genes for traits associated with seedling-stage alkalinity tolerance.

## 2. Results

### 2.1. Phenotypic Evaluation under Alkaline Stress

There was a wide variation in the alkalinity tolerance scores (AKT) in the diversity panel. The highly tolerant, tolerant, moderately tolerant, susceptible, and highly susceptible genotypes constituted 4%, 24%, 33%, 26%, and 13% of the panel, respectively (Figure 1). The performance of some of the lines with known tolerance levels to salt stress under alkaline stress is shown in Figure 2. All morphological and physiological traits varied widely after exposure to alkalinity stress at the seedling stage (Table 1). All traits except shoot fresh weight (FW) and shoot dry weight (DW) were normally distributed (Figure 3). Analysis of variance showed significant differences among genotypes for AKT, shoot length (SHL), root length (RTL), root-to-shoot ratio (RSR), inverse shoot fresh weight (inv_FW), log shoot dry weight (log_DW), shoot Na^+^ concentration (SNC), shoot K^+^ concentration (SKC), and shoot Na^+^: K^+^ ratio (SNK). AKT, SKC, and SNK showed high heritability (>80%), while medium heritability (50–80%) was observed for the rest of the traits (Table 1). 

No significant difference was observed among genotypes in the control experiment (Appendix A and Table 1). The mean AKT score was lower in the control compared to the stress environment. There were reductions in shoot and root length under alkalinity stress. The mean value of SNC and SNK was lower while the mean SKC value was higher under control than the stress environment. Similarly, heritability values were low for all traits under alkalinity stress.

### 2.2. Correlation Analysis

Significant correlations were observed among different traits (Table 2). AKT was negatively correlated with SHL, RTL, RSR, inv_FW, log_DW, and SKC, while it was positively correlated with SNC and SNK. SHL and RTL had a positive association with inv_FW, log_DW, and SKC; however, both traits were negatively correlated with RSR, SNC, and SNK. SKC had a significant positive correlation with all the morphological traits except AKT. There was a significant positive correlation between SNC and SNK. Inv_FW showed a positive correlation with log_DW and SKC but it was negatively correlated with SNC and SNK.

### 2.3. Principal Component Analysis (PCA) 

Nine morphological and physiological traits were used to separate the genotypes of the panel into different groups reflecting the level of tolerance to alkalinity stress. The first two principal components, PC1 and PC2, accounted for 38% and 22% of the total variation, respectively (Appendix A, Figure 4a). The first four principal components with eigenvalues greater than 1 explained 90% of the total variation. Based on a cutoff value of 0.50, four variables contributed to PC1 and PC2 and one variable contributed to PC3 (Appendix A). Different principal components might be responsible for explaining the variability for different sets of variables as there was no correlation among principal components. It was evident from the PCA plot that positively correlated traits were grouped together. PC1 accounted for the variability among rice genotypes for Inv_FW, log_DW, and SNC with their positive coefficients, and AKT with its negative coefficient (Appendix A). Similarly, PC2 represented the variation for RSR and SKC with their positive coefficients and STL and SNK with their negative coefficients. PC3 explained the variation among genotypes for RTL. PCA did not separate the *indica* and *japonica* genotypes based on phenotypic traits (Figure 4b).

### 2.4. Phenotypic Clustering

The genotypes of the panel were organized into five clusters based on the phenotypic response under alkaline stress (Table 3, Figure 5). The alkalinity tolerance level of each cluster was assessed by the mean AKT scores and mean SNC and SKC values. The genotypes grouped in cluster 1 were highly susceptible to alkalinity stress. The mean AKT, SNC, and SKC of the genotypes in cluster 1 were 8.1, 2056 mmol/kg, and 602 mmol/kg, respectively (Appendix A). This cluster included alkalinity-stress-susceptible genotypes N22 and Dular [2,7] and many genotypes from Arkansas, Texas, and Louisiana. The genotypes in cluster 2 were considered as tolerant to alkali stress. The alkalinity-tolerant genotypes Cocodrie [2,7], Jupiter, FL478, Bengal, and IR64 were placed in this group. The average AKT, SNC, and SKC for this cluster were 3.6, 1626 mmol/kg, and 629 mmol/kg, respectively. Most of the genotypes in this cluster were from Louisiana, California, India, and the Philippines. Cluster 3 was a highly tolerant group that included genotypes from Louisiana (JN100, JN349, W149) and genotypes from other countries (IR29, Vandana, Nipponbare). The mean AKT, SNC, and SKC were 1.8, 1487 mmol/kg, and 635 mmol/kg, respectively. Cluster 4 was moderately tolerant with mean AKT, SNC, and SKC of 5.6, 1793 mmol/kg, and 710 mmol/kg, respectively. Salt-tolerant genotypes Pokkali and TCCP were included in this group. Genotypes in cluster 5 were alkali-susceptible with mean AKT, SNC, and SKC of 7.0, 1956 mmol/kg, and 731 mmol/kg, respectively.

### 2.5. Population Structure

The population structure was determined using the Bayesian clustering method in the ‘STRUCTURE’ software. Five distinct groups (K = 5) were identified using the log-likelihood LnP (D) and Evanno’s deltaK (Figure 6). The list of genotypes, their geographic origin, and the subgroups are listed in Appendix A. Subgroup 1 (SG1) contained 40 genotypes belonging to the *japonica* subspecies. This subgroup contained genotypes from Louisiana, Arkansas, and Texas. The subgroups 2, 3, and 5 (SG2, SG3, and SG5) contained an admixture of *indica* and *japonica* subspecies. There was no distinction between US genotypes and those obtained from other countries in these subgroups. Nine genotypes were clustered in SG4, and all belonged to the *indica* subspecies. Furthermore, all genotypes in SG4 were genotypes from other countries except BHA1115, a black-hulled weedy rice genotype. The analysis of molecular variance (AMOVA) of the five subgroups showed that there were significant differences between and among these subgroups. The total variation among and within the subgroups was 61% and 39%, respectively (Table 4).

### 2.6. Linkage Disequilibrium (LD) 

A final set of 830 SNP markers was used for the GWAS analysis. The number of SNPs within a 1Mb window size is shown in Appendix A. The average SNP density ranged from 367 kb/marker for chromosome 6 to 537 kb/marker for chromosome 4, with an average of 450 kb/marker (Table 5). The mean LD decay over the physical distance, computed as *r*^2^, was 10,691 kb with a range of 7334 kb for chromosome 10 to 15,193 kb for chromosome 1. In the whole panel, the *r*^2^ estimate was 0.47 (Figure 7). The LD decay was faster for chromosomes 1, 2, 3, 4, 6, and 7 and was greater than its average value. The LD decay was slower for chromosomes 5, 8, 9, 10, 11, and 12. 

### 2.7. GWAS Analysis

GWAS was conducted via the FarmCPU model on all alkalinity-stress-related traits for 184 *indica* and *japonica* genotypes considering kinship (K) and population structure (Q) using the rMVP package in R. 31. Significant SNPs (*p* < 0.011) associated with AKT, SHL, RTL, RSR, Inv_FW, log_DW, SKC, SNC, and SNK were identified (Figure 8). There were 18 significant SNPs for morphological traits (AKT, SHL, RTL, RSR, log_DW, inv_FW) and 10 for physiological traits (SNC, SKC, SNK) (Table 6). 

### 2.8. Candidate Genes/QTLs for Alkalinity Tolerance

The significant SNPs were used to identify candidate genes present within the genes or present within 10 kb flanking genomic regions of the respective SNPs. A total of 28 candidate genes were detected (Table 6). These genes were compared with the earlier QTLs and differential gene expression studies [2,7,16]. Six genes were present within the intervals of earlier identified QTLs [2,7], while three genes were differentially expressed under alkaline stress in an earlier study [16]. 

Separately, a region was considered a QTL if more than two significant SNPs were present within the LD interval. Six large-impact QTLs were detected on chromosomes 1, 4, 7, 9, 10, and 12 (Table 7). Three QTLs (*qlog_DW1.37*, *qSKC9.19*, and *qSKC10.18*) were congruent to the QTLs identified previously [2,7]. The *qSNK4.34* co-localized with the *qSNK4-2* identified under alkalinity stress [12]. Two novel QTLs, *qSNC7.29* and *qSHL12.23*, were also identified on chromosomes 7 and 12, respectively.

### 2.9. Expression Profiling of Selected Candidate Genes under Alkalinity Stress 

Eight genotypes were selected from the tolerant (JN100, Cheniere, Cocodrie, Nipponbare) and susceptible (N22, Dular, Cypress, Hasawi) genotypes for expression analysis under alkalinity stress (Figure 9). The genes and primers used in the qRT-PCR analysis are listed in Appendix A. LOC_Os04g50090 (Helix–loop–helix DNA-binding protein) was downregulated in all the genotypes in the tolerant group compared with those in the susceptible group. The LOC_Os08g25390 (Bifunctional homoserine dehydrogenase) showed the same expression pattern in both groups except alkalinity-tolerant Cheniere. In contrast, the expression level of LOC_Os09g32972 (MYB protein), LOC_Os09g38340 (ZOS9-17—C2H2 zinc finger protein), and LOC_Os08g25480 (Cytochrome P450) increased 6 h after exposure to alkalinity stress in the tolerant group, whereas it decreased in all the genotypes in the susceptible group. A similar trend was observed for LOC_Os08g23440 (amino acid permease family protein) except for Cheniere, in which it was downregulated. There was downregulation in LOC_Os04g58160 (Fiber protein Fb34) in both groups with the exceptions of JN100 (tolerant) and Cypress (susceptible). Similarly, the expression of LOC_Os10g35230 (Rf1, mitochondrial precursor) decreased sharply after 6 h exposure to stress in both groups except Dular. 

## 3. Discussion

Salinity–alkalinity stress is a major hindrance to enhancing food production in many rice-growing areas around the globe [40]. Alkalinity stress can have harmful effects on plant growth and development. In addition to causing toxicity, it can also affect the stability and functioning of plant cells due to high pH [9]. While several studies reported the identification of QTLs and candidate genes for alkalinity-tolerance traits in rice [2,7,12,15,27,38], more research is needed to understand the molecular mechanisms. 

Alkaline tolerance evaluation in rice has been largely based on the uptake of sodium and potassium ions and the morpho-physiological response to alkalinity stress [2,12]. The alkalinity-tolerant plants sequester the Na^+^ outside the shoots and roots to tolerate high concentrations of Na^+^ around the rhizosphere [11]. This view was supported by our observation of a significant positive correlation between AKT and SNC (Table 2). In addition, an excess of sodium in the shoots indirectly affecting the upward movement of potassium in the plant was corroborated by a significant negative correlation between SNC and SKC (Table 2). The tolerant lines accumulate more K^+^ than the susceptible lines. The detrimental effects of alkalinity stress were clearly reflected in the negative correlation between AKT and morphological traits. The negative association between RTL and AKT, SNC, and SNK indicated that plants with deeper root systems could maintain a desirable Na:K ratio and uptake of essential nutrients such as Fe under a stress environment, as reported earlier [23]. The range of AKT and other traits was wide under alkaline stress (Figure 1, Figure 2 and Figure 3) and both *indica* and *japonica* genotypes included both tolerant and susceptible genotypes. IR29 (*indica*) and Cocodrie (*japonica*) were tolerant to alkaline stress, which led us to conclude that *indica* and *japonica* genotypes could not be distinguished based on alkalinity tolerance (Figure 2, Table 3). Increased mean AKT score and reduced heritability values for all traits under a stress environment compared to control (Table 1 and Appendix A) suggested the influence of alkaline stress on the expression of morpho-physiological traits.

There was a wide range of variability for alkalinity tolerance among the rice genotypes (Figure 4). Principle component analysis revealed that AKT, log_DW, and inv_FW had the highest contribution for the variation among the genotypes, while SNC, SNK, and RSR had a moderate contribution (Figure 4, Appendix A). These three traits could be used to assess the level of alkalinity tolerance in rice genotypes. De Leon et al. (2015) [41] used cluster analysis and multivariate test statistics to differentiate salt-tolerant rice genotypes from salt-sensitive genotypes based on morphological traits. Chunthaburee et al. (2016) [42] reported a strong correlation between Na^+^/K^+^ and salinity tolerance in rice and used this ratio to group genotypes into tolerant and sensitive groups. The strong positive correlation of AKT with SNC and SNK, and negative correlation between SKC and AKT, in our study confirmed that regulation of the uptake of Na^+^ and K^+^ is critical for plants’ survival under alkalinity stress. 

The rice genotypes were classified based on their level of tolerance to alkalinity stress. The salt-susceptible IR 29 [43] and Cheniere [44] were placed in the highly tolerant category (Table 3, Figure 5). The other genotypes in the tolerant group included both *japonica* (Jupiter, Bengal) and *indica* (FL478, Geumgangbyeo) genotypes. In an earlier study [41], Jupiter and Bengal were classified as susceptible and FL478 and Geumgangbyeo as tolerant to saline stress. The classification of Cocodrie as tolerant and N22 and Dular as highly susceptible in this study was consistent with earlier studies [2,7]. The inclusion of saline-tolerant Hasawi [45] and salt-susceptible Cypress [41] in the highly susceptible group suggested different mechanisms underlying tolerance to alkaline and saline stresses. In contrast, the grouping of some of the saline-tolerant genotypes (Pokkali, Nona Bokra, TCCP, Nipponbare, Geumgangbyeo, and Damodar) in the highly tolerant, moderately tolerant, and tolerant groups suggested that there might be some common physiological responses and the co-expression of genes under both types of stress. 

Population structure analysis identified five subgroups within the *indica* and *japonica* rice genotypes (Figure 6b). Three subgroups were earlier detected within the *japonica* subgroup under alkaline stress [12]. This inconsistency could be due to the inclusion of the *indica* subgroup in our study. There was some clustering of *indica* and *japonica* genotypes, with subgroup 1 (SG1) containing all *japonica* genotypes and SG4 containing all the *indica* genotypes (Appendix A). However, population structure analysis did not show an extreme differentiation among US genotypes or between *indica* and *japonica* genotypes. This observation was in disagreement with an earlier study [46], which showed the distinct separation of US and Asian rice genotypes. 

Alkaline stress results in osmotic stress, ionic imbalance, and nutritional deficiency due to high pH [47,48,49], and, therefore, is difficult to mitigate. Studies have demonstrated that the expression of stress-responsive genes under stress can enhance the tolerance of plants to various abiotic stresses [50,51,52]. Multiple strategies have been utilized to identify genes related to alkaline tolerance in rice, and several genes have been identified [30,31,33]. 

Twenty-eight SNPs significantly associated with alkalinity tolerance suggested the involvement of 28 different loci (Table 6). Among these significant SNPs, at least two were associated with all traits except inv_FW. The SNPs identified in this study were compared with the QTLs and candidate genes for alkalinity tolerance in rice. Seventeen SNPs overlapped with QTLs and genes from earlier studies (Table 6). Based on LD decay, we identified a QTL, *qSNK4*, in the same region with a 5 Mb interval and R^2^ value of 16% (Table 7). Some known Na^+^ and K^+^ transporters such as *OsHKT1;1* [53], *OsHKT1;4* [54], and *OsHAK15* [55] were present within this interval. The role of Na^+^ and K^+^ transporters in improving abiotic stress tolerance is well known [56,57,58]. It is possible that these proteins enhance alkalinity tolerance by regulating the uptake of Na^+^ ions and maintaining the Na^+^:K^+^ ratio [53]. Two significant SNPs (S04_29715617 and S04_34925111) for RTL and one SNP each for SNK (S04_34643455) and AKT (S04_29881066) on chromosome 4 tagged to LOC_Os04g4950, LOC_Os04g58730, LOC_Os04g58160, and LOC_Os04g50090, respectively, were located in the same region as *qSNK4-2* [12]. The role of *HKT* and *HAK* genes in the regulation of Na^+^ and K^+^ under high pH conditions has remained elusive to date warranting investigation of this genomic region on chromosome 4 in the future to provide some insights into alkalinity tolerance mechanisms in rice. We included two genes, LOC_Os04g58160 (Fiber protein Fb34, putative) and LOC_Os04g50090 (Helix–loop–helix DNA-binding protein) from this genomic region in the qRT-PCR analysis. There was no clear contrast in the expression pattern of LOC_Os04g58160 between the tolerant and susceptible groups (Figure 9), but LOC_Os04g50090 was downregulated in all genotypes in the tolerant group compared to the susceptible group. These findings suggest the involvement of the novel gene LOC_Os04g50090, which played a negative role in regulating alkalinity tolerance in rice. Several members of the bHLH family were reported to be responsive to abiotic stresses in multiple species [59,60,61,62]. 

Similarly, two significant SNPs each on chromosomes 9 and 10 led to the detection of *qSNC9* and *qSKC10*, which co-localized with the *qSNC9.19* and *qSKC10.18* from earlier studies (Table 7) [2,7]. Based on LD decay, *qSKC10* and *qSKC9* spanned over 735 and 361 kb region, respectively. Six candidate genes were differentially expressed between alkalinity-tolerant and -susceptible genotypes in our previous study [2]. Furthermore, these genes were involved in the abiotic stress tolerance response in rice [29,63,64,65,66,67,68,69,70]. The genes harboring significant SNPs identified in this study were LOC_Os09g32350 (Expressed protein), LOC_Os09g32972 (MYB protein), and LOC_Os10g35230 (Rf1, mitochondrial precursor). These genomic regions can provide a deeper understanding of the molecular basis of alkalinity stress tolerance. Finally, creating mutations using CRISPR-mediated genome editing followed by analyzing the phenotypic changes can help validate the functional role of these genes in response to alkalinity stress. A genome-wide comparative analysis of the MYB gene family in rice and *Arabidopsis* implicated several MYB proteins in abiotic stress responses [71]. The upregulation of LOC_Os09g32972 in tolerant genotypes suggests a crucial role for the MYB transcription factor in alkalinity tolerance in rice (Figure 9). However, there was no clear trend in the expression of LOC_Os01g35230 (Rf1, mitochondrial precursor), which was downregulated in both tolerant and susceptible genotypes in this study.

The *qlog_DW1.38* detected in this study was the same QTL identified previously [72,73]. The *sd1* locus (LOC_Os01g66100) [74] was located within this interval. De Leon et al. (2016) [72] showed that the *sd1* gene was responsible for increasing SRR in plants under saline stress. The same region was associated with various morphological traits under alkalinity stress [2,7], suggesting the role of this region in alkalinity tolerance. Two QTLs, *qSNC7* and *qSHL12*, were considered to be novel QTLs. The candidate genes linked to significant SNPs detected within the *qSNC7* were histone-arginine methyl transferase *CARM1* (LOC_Os07g47500) and protein kinase *APK1B* (LOC_Os07g49470) (Table 6 and Table 7). Similarly, a protein kinase-like family protein (LOC_Os12g37570) and xaa-pro aminopeptidase (LOC_Os12g37640) were present within the *qSHL12*. A calcium-dependent protein kinase *OsDMI3* was earlier shown to enhance the ability of rice roots to tolerate saline–alkaline conditions by regulating the intake of Na^+^ and H^+^ ions [32]. Similarly, histone posttranslational modifications (PTMs) and interactions between them play a key role in mediating salt tolerance in plants [75,76]. Our results identified potential genetic targets for improving the growth of rice in environments with high alkalinity levels. Three significant SNPs tagged with candidate genes, amino acid permease family protein (LOC_Os08g23440), Cytochrome P450 (LOC_Os03g25480), and bifunctional homoserine dehydrogenase (LOC_Os08g25390), co-localized with the three earlier reported differentially expressed genes under alkaline stress [16] (Table 6). Among these genes, LOC_Os08g23440 and LOC_Os03g25480 were upregulated in the tolerant group, whereas LOC_Os08g25390 was downregulated in the tolerant group (Figure 9). The amino acid permease1, which is induced by salt stress, mediated the uptake of proline, and increased salt susceptibility was observed in an *Arabidopsis* mutant due to reduced accumulation of proline compared with the wild type [77]. The upregulation of this gene in tolerant genotypes suggested its role in imparting alkalinity tolerance in rice. Cytochrome P450 was known to confer abiotic stress tolerance [78,79]. Since the role of bifunctional homoserine dehydrogenase under abiotic stresses is unclear, it warrants future investigation for its potential in improving alkaline stress tolerance in rice. 

The other candidate genes tagged with the significant SNPs were ER-Golgi intermediate compartment protein 3 (LOC_Os04g38340), *OsSIGP1* (LOC_Os02g58139), anthocyanidin 5,3-O-glucosyltransferase (LOC_Os01g64910), *OsFBX168* (LOC_Os05g41130), Cysteine protease family protein (LOC_Os02g06890), C2H2 zinc finger protein (LOC_Os09g38340), and fiber protein Fb34 (LOC_Os04g58160) (Table 6). LOC_Os04g38340, LOC_Os02g58139, LOC_Os01g64910, and LOC_Os04g58160 were not reported in previous alkalinity tolerance studies. However, fiber protein Fb34 did not show a contrasting expression pattern between the tolerant and susceptible groups in our study (Figure 9). The expression of C2H2 zinc finger protein sharply increased in all tolerant genotypes compared with the susceptible group (Figure 9). The F-box protein, cysteine protease family protein, and C2H2 zinc finger protein were involved in abiotic stress tolerance [2,7,80]. The C2H2 zinc finger protein is a novel candidate gene for alkalinity tolerance identified in this study. 

Several studies identified the same candidate genes and QTLs for highly correlated traits [81,82]. However, this study revealed different results and was in agreement with earlier studies [83,84]. Although a high correlation was observed between Fe and Zn content, different QTLs controlled these traits [83]. Similarly, different candidate genes were identified for the highly correlated traits, grain width and grain weight [84]. Our results could be due to the complexity of alkalinity tolerance mechanisms in which associated phenotypic traits are controlled by many genes and environmental conditions. Another reason could be the low SNP density, which resulted in a failure to detect more SNPs significantly associated with alkalinity tolerance traits. 

Given the importance of rice as a staple food and the challenges due to climate-change-related environmental stresses, and geographical limitations, it is crucial to characterize the response of different rice genotypes to alkalinity stress. This study revealed significant variations in the physiological responses of different rice genotypes to alkalinity stress. Specifically, we observed that increased alkalinity led to increased uptake of Na^+^ in shoots, which in turn resulted in decreased uptake of K^+^. The study identified several alkalinity-tolerant rice genotypes such as Saturn, Della, JN100, JN349, Nipponbare, Mercury, BHA1115, IR 29, Cheniere, IR 50, Panidhan II, Koshihikari, Teqing, Dellmont, Kanchan, Swarna, W149, Perum karuppan, Taipe 309, and Hayamasari which could be used as potential donors to improve alkalinity tolerance. The high-yielding tolerant genotypes could be also recommended for cultivation in saline–alkaline areas. Most of these top-performing genotypes had relatively low-to-moderate Na^+^ uptake while maintaining a high K^+^ uptake indicating a higher degree of tissue tolerance. In addition to the QTLs *qSNC9* and *qSKC10*, which co-localized with previously reported QTLs, a novel QTL and several candidate genes were identified. 

## 4. Conclusions

Alkalinity tolerance exhibited by salt-susceptible rice genotypes, or vice versa, indicated that different mechanisms underly tolerance to salinity and alkalinity stresses. Twenty-eight significant SNPs and six QTL regions were detected via GWAS analysis. Two QTLs, *qSNC9* and *qSKC10*, were congruent with the QTLs detected in previous alkaline studies, while *qSNC7* was a novel QTL. Six candidate genes, LOC_Os04g50090, LOC_Os08g23440, LOC_Os09g32972, LOC_Os08g25480, LOC_Os08g25390, and LOC_Os09g38340, showed contrasting expression between tolerant and susceptible genotypes, suggesting their potential role in imparting alkalinity tolerance. Since the understanding of alkalinity tolerance in rice is limited, the alkaline-tolerant genotypes, SNPs, QTLs, and candidate genes identified in this study will be valuable resources for gaining further insights into the tolerance mechanism, as well as for breeding alkaline-tolerant rice varieties. 

## 5. Materials and Methods

### 5.1. Plant Materials

A total of 185 rice genotypes from the USA and the International Rice Research Institute (IRRI) including some saline-alkaline tolerant genotypes were evaluated under alkalinity stress at the seedling stage. The experiment was conducted in a randomized complete block design (RCBD) at the LSU AgCenter greenhouse with three replications. Seeds were exposed to 50 °C for 3 days to break dormancy and sown in 4-inch sand-filled pots. The seedlings at the two-leaf stage were then exposed to alkalinity stress for two weeks with 0.20% Na_2_CO_3_, followed by one week of exposure to 0.40% Na_2_CO_3_ in the stress experiment. The pH of the solution was adjusted to 10.0 in the stress experiment to imitate alkaline stress. The seedlings in the control experiment were allowed to grow under normal conditions. After three weeks of stress, the seedlings were evaluated for various morphological and physiological traits. The mean performance of each genotype was recorded for alkalinity tolerance score (AKT), root length (RTL), shoot length (SHL), root-to-shoot ratio (RSR), shoot dry weight (DW), shoot fresh weight (FW), shoot Na^+^ concentration (SNC), shoot K^+^ concentration (SKC), and shoot sodium-to-potassium ratio (SNK). Alkalinity tolerance scoring was done on a scale of 1–9 depending on the percentage of dry and yellow leaves [2]. The shoot length and root length of the seedlings were recorded. The fresh weight of the shoots was recorded and the dry weight of the shoots was recorded after drying the seedlings at 50 °C for one week. The shoot samples were harvested and oven-dried at 60 °C for 10 days and 0.1 g of homogenized sample from each genotype was digested with nitric acid: hydrogen peroxide (5:3 mL) at 152–155 °C for 3 h [85]. The Na^+^ and K^+^ content of the samples was measured using a flame photometer (Jenway model PFP7, Bibby Scientific Ltd., Staffordshire, UK). The final concentrations of Na^+^ and K^+^ were calculated from the standard curve derived from standard solutions of Na^+^ and K^+^. The ratio of shoot Na^+^ to K^+^ (SNK) was obtained by dividing shoot Na^+^ concentration by shoot K^+^ concentration.

### 5.2. Statistical Analysis 

SAS version 9.4 [86] and R version 2.2.1 [87] were used for statistical analysis of phenotypic data. The Shapiro–Wilk test was performed to test the hypothesis of normality and the normality assumption was violated in the case of DW and FW. Log and inverse transformations were performed for the data on DW and FW, respectively. Histograms and descriptive statistics were obtained. For each trait, an analysis of variance (ANOVA) was performed. Pearson correlation coefficients were computed to better understand the association among traits for alkalinity tolerance. The mean value of each trait was used for clustering and PCA. Principal component analysis was done to investigate the relationship among *indica* and *japonica* genotypes and the contribution of variables to the phenotype. The clustering of the genotypes was done using Euclidean distance. Broad sense heritability was calculated in SAS following the method of Holland et al. [88].

### 5.3. SNP Genotyping and Quality Control

Leaf samples of each genotype were collected from 21-day-old seedlings. The leaf samples were sent to the AgriPlex Genomics (https://www.agriplexgenomics.com/ Accessed on 15 August 2022) sequencing facility for genotyping with the IRRI rice amplicon SNP panel. SNP calling was done by Agriplex Genomics. SNPs with minor allele frequency (MAF) < 0.05 and SNPs with >5% missing values were filtered out. The k-nearest neighbor (KNN) method was used for the imputation of the missing values [89]. After filtering, a total of 832 SNPs were used for the analysis.

### 5.4. Structure Analysis and Linkage Disequilibrium

To assess the population structure and assign individuals to populations, STRUCTURE 2.3.4 [90] was used. The structure with varying numbers of groups (K) from 1 to 10 was run with 10,000 burn-in-periods and 50,000 Markov chain Monte Carlo (MCMC) replications. The ad hoc ΔK statistic was used to determine the true value of K [91]. 

Pairwise linkage disequilibrium was computed in Tassel 5.0 software using squared correlation coefficients (*r*^2^) of alleles [92]. LD decay was estimated in R using non-linear curves [93]. The rate of LD decay was computed as the physical distance between markers.

### 5.5. Association Mapping

Population stratification often results in false positives in the GWAS analysis. To effectively control these false positives using population structure and kinship matrix, several GLM [94,95] and MLM [94,96] methods can be used. These methods compromise true positives in order to control false positives. The fixed and random model circulating probability unification (FarmCPU) method can reduce overfitting and effectively control false positives without compromising true positives [97]. Given those advantages, genome-wide association mapping was conducted via the FarmCPU method in the R package. The fixed effect model (FEM) and the random effect model (REM), which are used iteratively, are the two components of the multiple loci linear mixed model in FarmCPU. For the prevention of overfitting, REM estimates the several associated markers to calculate kinship, while FEM controls false positives and negatives by testing markers one at a time and kinship from REM as covariates [97]. The *p*-values of testing markers and various associated markers were unified at each iteration. The mean values of nine physiological and morphological traits were used for genome-wide association analysis. After GWAS, a threshold value (−log_10_ (*p*) ≥ 1.95) equivalent to 0.011 was used for declaring significant SNPs.

### 5.6. Candidate Gene Analysis

Candidate genes associated with significant SNPs (*p* ≤ 0.001) were selected if the genes were present within ±10 kb of the significant SNPs in the reference rice genome sequence. The functional annotation of candidate genes was evaluated. The list of candidate genes from GWAS analysis was compared with the QTLs and differential expression genes under alkalinity stress in rice from previous studies.

### 5.7. Expression Profiling of Selected Genes by Real-Time Quantitative Reverse Transcription PCR (qRT-PCR)

Four genotypes each from the tolerant and susceptible groups were selected for gene expression profiling. These genotypes were JN100, Cheniere, Cocodrie, and Nipponbare from the tolerant group and N22, Dular, Cypress, and Hasawi from the susceptible group. A hydroponic setup containing 1 g/L of Jack’s professional fertilizer (20-20-20) (JR Peters Inc., Allentown, PA, USA) was used to grow seedlings of these genotypes. The experiment was conducted in a randomized complete block design (RCBD) at an LSU greenhouse. There were three replications each for the control and alkalinity stress experiments. Twenty to thirty seedlings of each genotype were grown in each replication. Seedlings in the stress experiment were exposed to alkaline stress at the two-leaf stage with a solution of 0.5% Na_2_CO_3_ and pH 10.0. Leaf samples were collected from the control and stress experiments at 0 and 6 h of stress exposure and were stored at −80 °C. Total RNA was isolated from leaf tissues using Trizol reagent ((Thermofisher Scientific, Waltham, MA, USA). The quality of the RNA was assessed using a 1.2% agarose gel and the quantity was determined using an ND-1000 spectrophotometer (Thermofisher Scientific, Waltham, MA, USA). The RNA samples were treated with PerfeCTa DNase1 (Quantabio, Beverly, MA, USA) to remove any contaminating genomic DNA, and cDNA was synthesized using iScript™ first strand cDNA synthesis kit (Bio-Rad Laboratories, Hercules, CA, USA). Primers were designed using the PrimerQuest Tool (Integrated DNA Technologies, Inc., Coralville, IA, USA) and *EF1α* was used as an internal control. The qRT-PCR reaction was performed in triplicate using pooled cDNA from the biological replicates [98]. The expression level of genes was determined using the 2^–∆∆CT^ method [99], which involves normalization to the internal control gene *EF1α* and calculation of the fold change in expression under alkalinity stress compared to the control for each genotype.

## Figures and Tables

**Figure 1 plants-12-02206-f001:**
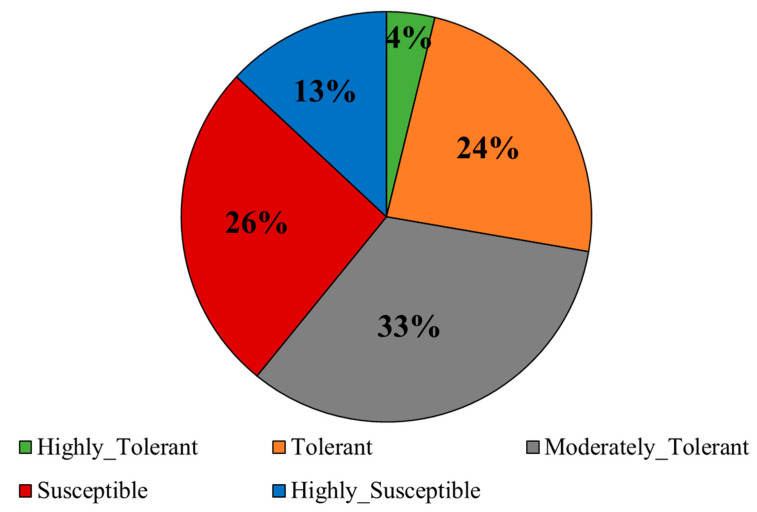
Variation in alkalinity tolerance among the rice genotypes based on alkalinity tolerance score (AKT) at the seedling stage.

**Figure 2 plants-12-02206-f002:**
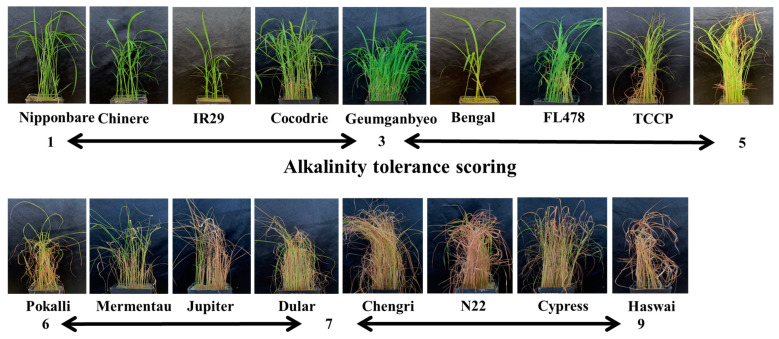
Performance of some of the lines (known salt-tolerance level) after 21 days of alkaline stress at the seedling stage. The scale represents the alkalinity tolerance score (AKT) after the stress on a scale of 1 (highly tolerant) to 9 (highly susceptible).

**Figure 3 plants-12-02206-f003:**
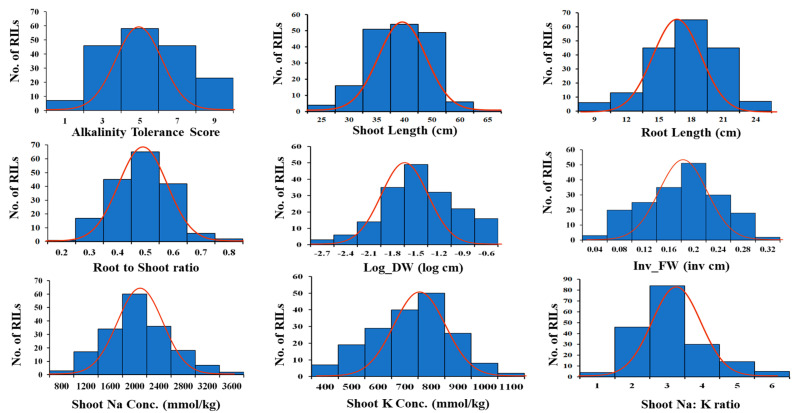
Frequency distribution of nine morphological and physiological traits under alkaline stress. AKT, alkalinity tolerance score; SHL, shoot length; RTL, root length; RSR, root-to-shoot ratio; inv_FW, inverse fresh weight; log_DW, log dry weight; SNC, shoot Na^+^ concentration; SKC, shoot K^+^ concentration; SNK, shoot Na^+^:K^+^ concentration.

**Figure 4 plants-12-02206-f004:**
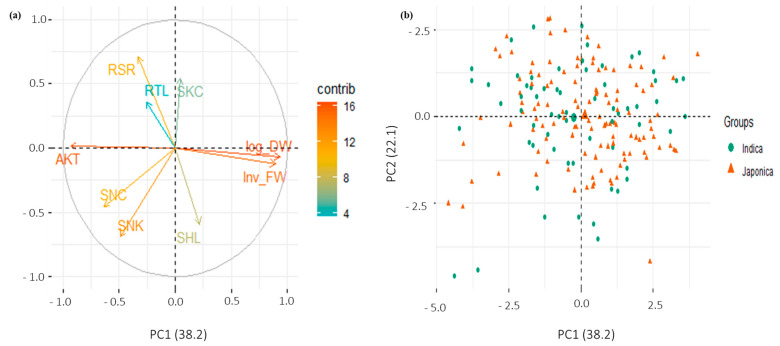
Principal component analysis (PCA) plot. (**a**) Grouping of variables associated with nine morphological and physiological traits of rice genotypes under alkalinity stress at the seedling stage, (**b**) Scatter plot of the *indica* and *japonica* rice genotypes represented in the two major principal component axes. No sufficient clustering was observed between *indica* and *japonica* genotypes. AKT, alkalinity tolerance score; CHL, chlorophyll content (SPAD units); SHL, shoot length (cm); RTL, root length (cm); log_DW, log dry weight (gm); inv_FW, inverse fresh weight (gm); SNC, Shoot Na^+^ concentration (mmol/kg); SKC, shoot K^+^ concentration (mmol/kg); SNK, shoot Na^+^:K^+^ ratio.

**Figure 5 plants-12-02206-f005:**
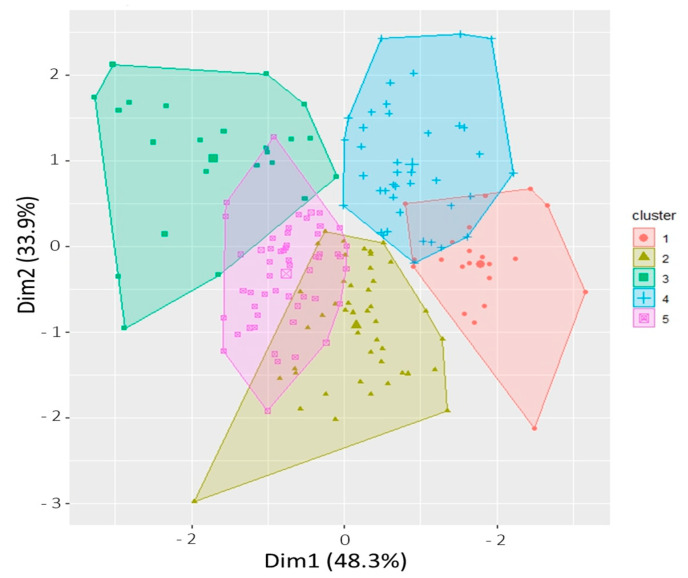
Phenotypic clustering of rice genotypes by UPGMA based on Euclidean distance computed from nine morphological and physiological traits under alkalinity stress at the seedling stage.

**Figure 6 plants-12-02206-f006:**
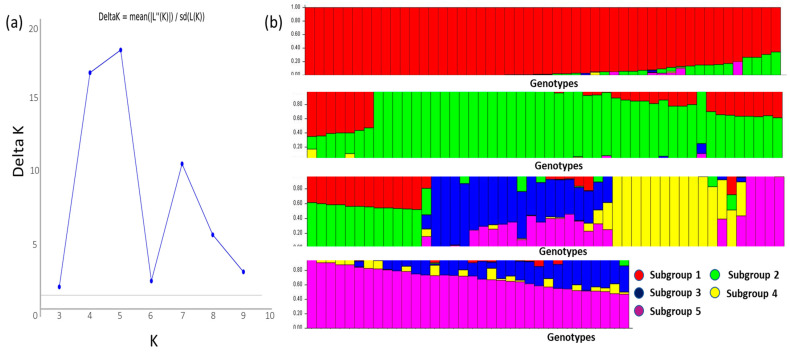
Population structure analysis of rice genotypes. (**a**) identification of the optimum number of subpopulations using LnP(D) derived ΔK. The maximum value of ΔK was found to be at K = 5, suggesting a division of the entire population into five subpopulations. The X-axis shows the number of subgroups (K) and Y-axis shows rate change of log probability values (ΔK) with change in K (**b**) Assignment of rice genotypes into five subpopulations, with the X-axis and Y-axis representing genotypes and the proportion of genetic ancestry in the subgroup membership, respectively. The genotypes present in each subgroup are listed in Appendix A.

**Figure 7 plants-12-02206-f007:**
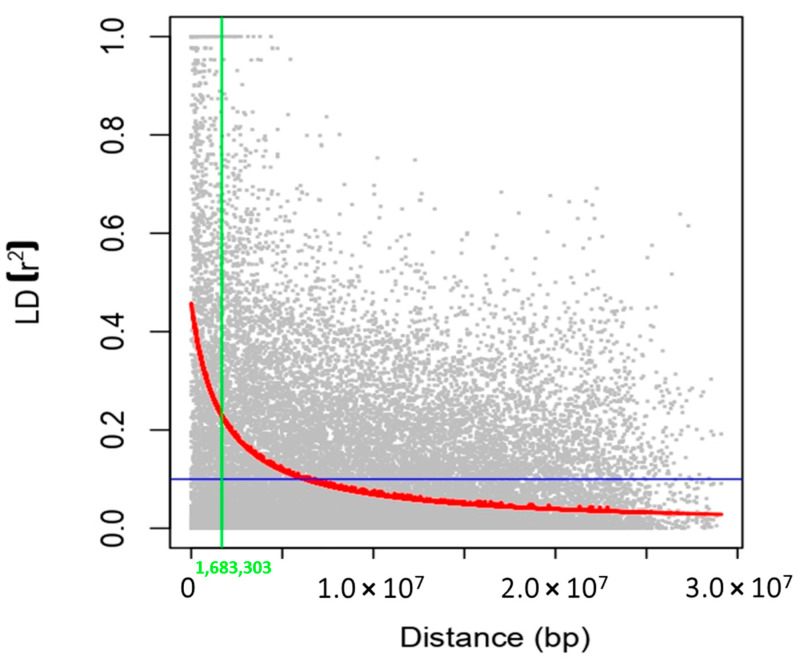
Genome-wide average linkage disequilibrium decay across all chromosomes. The X-axis and Y-axis represent the distance (bp) and LD, respectively. The intersection of green and blue lines indicates the derived threshold for LD due to linkage at respective distance (blue line) and LD (green line).

**Figure 8 plants-12-02206-f008:**
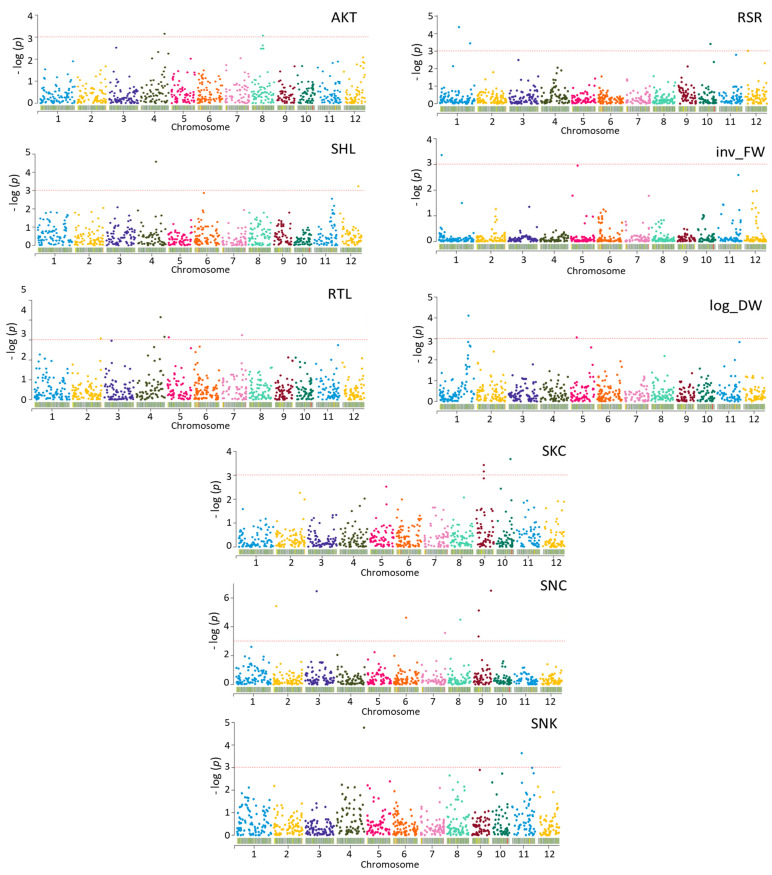
Manhattan plots of the markers associated with alkalinity tolerance in rice. The X-axis shows markers along the 12 rice chromosomes and the Y-axis shows the negative log_10_- transformed *p*-values for each association. Red dotted lines indicate the significance threshold. AKT, alkalinity tolerance score; SHL, shoot length; RTL, root length; RSR, root-to-shoot ratio; inv_FW, inverse fresh weight; log_DW, log dry weight; SNC, shoot Na^+^ concentration; SKC, shoot K^+^ concentration; SNK, shoot Na^+^:K^+^ ratio.

**Figure 9 plants-12-02206-f009:**
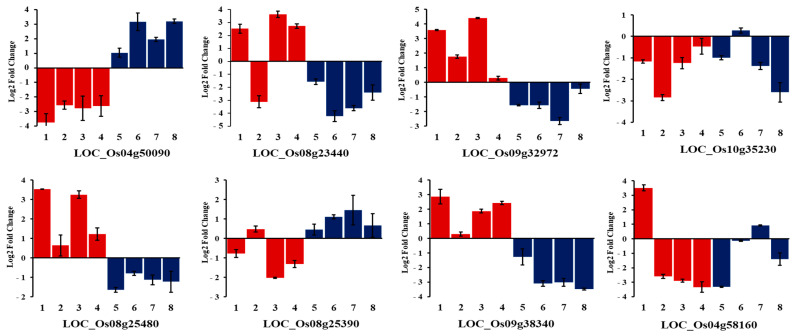
Expression profiles of selected genes present under alkalinity stress (6 h after imposition of stress) in the tolerant and susceptible groups. Red and blue in the bars represent tolerant and susceptible groups, respectively. Genotypes included in the experiment were: 1—JN100; 2—Cheniere, 3—Cocodrie; 4—Nipponbare; 5—N22; 6—Dular; 7—Cypress; 8—Hasawi. EF1α was used as the reference gene and gene expressions were calculated as log2-fold changes under alkaline stress compared with control in all genotypes. LOC_Os04g50090—Helix–loop–helix DNA-binding protein; LOC_Os08g23440—amino acid permease family protein; LOC_Os09g32972—MYB protein; LOC_Os10g35230—Rf1, mitochondrial precursor; LOC_Os03g25480—cytochrome P450; LOC_Os08g25390—Bifunctional homoserine dehydrogenase; LOC_Os09g38340—ZOS9-17—C2H2 zinc finger protein, LOC_Os04g58160—Fiber protein Fb34, putative.

**Table 1 plants-12-02206-t001:** Phenotypic performance of rice genotypes under alkalinity stress at the seedling stage.

Trait ^a^	Min	Max	Mean	Standard Deviation	RIL Pr > Fc ^b^	Heritability
AKT	1.0	9.0	4.75	2.05	0.002 **	0.87
SHL	21.3	63.0	37.7	6.6	0.047 *	0.64
RTL	7.0	22.7	16.4	3.2	0.029 *	0.52
RSR	0.16	0.76	0.46	0.21	0.029 *	0.69
Inv_FW	0.01	0.30	0.16	0.07	0.0003 **	0.61
log_DW	−2.84	−0.60	−1.50	0.43	0.029 *	0.53
SNC	661.6	3508.1	1730.1	440.3	0.032 *	0.77
SKC	326.3	1077.9	657.7	138.3	0.048 *	0.84
SNK	0.88	5.95	2.79	1.11	0.041 *	0.81

^a^ AKT, alkalinity tolerance score; SHL, shoot length; RTL, root length; RSR, root-to-shoot ratio; inv_FW, inverse fresh weight; log_DW, log dry weight; SNC, shoot Na^+^ concentration; SKC, shoot K^+^ concentration; SNK, shoot Na^+^:K^+^ ratio. ^b^ Genotypic difference among lines; *, ** significant differences between the means of genotypes at 0.05 and 0.01 level of probability, respectively; Fc—Analysis of variance test.

**Table 2 plants-12-02206-t002:** Pearson correlation coefficients between morphological and physiological traits in rice genotypes under alkalinity stress.

Trait ^a^	AKT	SHL	RTL	RSR	Inv_FW	log_DW	SNC	SKC	SNK
AKT	1.000								
SHL	−0.123 *	1.000							
RTL	−0.147 *	0.02	1.000						
RSR	−0.154 *	−0.757 **	−0.533 **	1.000					
Inv_FW	−0.909 **	0.127 *	0.174 *	−0.173 *	1.000				
log_DW	−0.954 **	0.139 *	0.170 *	−0.181 *	0.961 **	1.000			
SNC	0.467 **	−0.028	−0.074	−0.027	−0.36 **	−0.423 **	1.000		
SKC	−0.053 *	0.004	0.009	0.002	0.155 *	0.094	−0.87 **	1.000	
SNK	0.321 **	−0.028	−0.002	−0.015	−0.18 **	−0.258 **	0.742 **	−0.674 **	1.000

^a^ KT, alkalinity tolerance score; CHL, chlorophyll content; SHL, shoot length; RTL, root length; RSR, root-to-shoot ratio; inv_FW, inverse shoot fresh weight; log_DW, log shoot dry weight; SNC, shoot Na^+^ concentration; SKC, shoot K^+^ concentration; SNK, shoot Na^+^:K^+^ ratio. * Significant at 0.05 level of probability; ** Significant at 0.01 level of probability.

**Table 3 plants-12-02206-t003:** Classification of rice genotypes based on various morphological and physiological traits under alkalinity stress.

Clusters	Genotypes
Cluster 1 (Highly Susceptible)	Hasawi, Roy J, Djogolan, Dular, Cypress, Vegold, ChN1264, Toro-2, Belle Patna, N22, Magnolia, Glutinous Zenith, Jazzman-2, Toro, Chengri, Azucena, Chambal, Bluebonnet, Orion, Adair, Pratao Tipo Guedes, Dholamon 560, Hill medium, KN-1-B-361-1-8-67
Cluster 2 (Tolerant)	PSBRC-50, CL111, Caloro, Cheriviruppu, CL131, Trenasse, Pirogue, LA0802140, Jupiter, LA0702085, Rexona, FL478, Geumgangbyeo, Neptune, CL261, FL318, Caffey, Lacassine, CLPK873, Cocodrie, Lacrosse, Sunbonnet, Lafitte, Dellmati, Carolina Gold, Bengal, Century Patna, CL152, Nato, MS-1996-9, Glutinous Selection, Saturn Rogue, Langmanbi, Milagrosa, Zhenshan 97, R-50, Mars, Kasalath, Sarioo50, IR 8, M202, Zenith, IR 64, Arkansas Fortuna, Texmont, Kranti, TP 49, Millie, Kirak, Chung yuen, IRGC1244, Newrex, RD, IRGC32567, Kitaake, Brazos, M-204, Delitus, Italica Livorno, CT-329
Cluster 3 (Highly Tolerant)	Saturn, Della, JN100, Moroberekan, JN349, Nipponbare, Mercury, BHA1115, IR 29, Dellrose, Lotus, Agami, Neches, Epagri, Cheniere, CSR11, Vandana, Gu Ze, IR 50, Panidhan II, Koshihikari, Teqing, Taichung 65, Daido, Lemont, Quilloa 66304, Dellmont, Kanchan, Swarna, W149, Perum karuppan, Taipe 309, Hayamasari
Cluster 4 (Moderately Tolerant)	LA110, CL142, Century Rogue, Pokkali, Pecos, Nona Bokra, Wells, Gold Zenith, Skybonnet, Tebonnet, Nira, Vista, TCCP, Templeton, Nova 66, IRRI147, Taggert, Bluebelle, Arang, Ecrevisse, Smooth Zenith, Damodar, Kalia, MS-1995-15, SLO16, Rexark, V20B, Ning Yang Keng, Stormproof, Starbonnet, B573-A4-20-6, R-27, Gold Nato, Naylamp, Azaurel, Melrose, Jinheung, Arkrose, Dixiebelle, Nerretto, PSRR-1, Bala, Co39, San Tou Thou, IR4432-52-6-4, Hill LongGrain, Bharathy, H4, IARI 5823, Early Prolific, Fatehpur 3, Prelude, WC10380
Cluster 5 (Susceptible)	Pinkaeo, LAH10, Mermentau, Evangeline, CR5272, R609, Jes, Della-2, CL162, R-54, Radin Ebos 33, Kokubelle, LaGrue, Jackson

**Table 4 plants-12-02206-t004:** Analysis of molecular variance (AMOVA) among the five subpopulations identified by ‘STRUCTURE’ software.

Source of Variation	DF ^a^	Sum of Squares	Mean Sum of Squares	Variance (%)	*p*-Value ^b^
Among population	4	495.8	123.9	61	<0.0001
Within population	163	1129.8	6.9	39	<0.001
Total	167	1625.6		100	

^a^ Degrees of freedom; ^b^ Level of significance.

**Table 5 plants-12-02206-t005:** Analysis of genome-wide linkage disequilibrium (LD) decay used for GWAS in this study.

Chr.	No. of SNPs	Chr. Size (bp) ^†^		SNP Density (bp/SNP)	LD ^$^ Distance (bp)
1	93	43,270,923		465,279	15,193,454
2	81	35,937,250		443,670	13,604,743
3	78	36,413,819		466,844	12,513,847
4	66	35,502,694		537,920	11,534,383
5	64	29,958,434		468,101	10,451,492
6	85	31,248,787		367,633	11,193,100
7	59	29,697,621		503,350	11,473,349
8	72	28,443,022		395,042	9,654,739
9	58	23,012,720		396,772	7,434,300
10	54	23,207,287		429,765	7,334,850
11	59	29,021,106		491,884	9,261,105
12	62	27,531,856		444,063	8,644,012
Total	830	373,245,519	Mean	450,861	10,691,115

^†^ According to Kawahara et al., 2013; **^$^** LD, Linkage disequilibrium.

**Table 6 plants-12-02206-t006:** Significant SNPs, associated genes, and co-localized QTLs or genes for alkalinity tolerance in this genome-wide association study.

Trait ^a^	SNP	Locus	Annotation	QTLs/Genes in Previous Studies
AKT	S04_29881066	Os04g50090	Helix–loop–helix DNA-binding protein	*qSNK4-2* [12]
	S08_14184612	Os08g23440	amino acid permease family protein	LOC_Os08g23440 [16]
SHL	S04_22808095	Os04g38340	ER-Golgi intermediate-compartment protein 3	*qDLR4* [26]
	S12_23066809	Os12g37570	protein kinase family protein	
	S12_23108164	Os12g37640	xaa-Pro aminopeptidase	
RTL	S02_35216781	Os02g58139	OsSigP1-Type I Signal Peptidase homolog	
	S04_29715617	Os04g49850	Expressed protein	*qSNK4-2* [12]
	S04_34925111	Os04g58730	AT-hook-motif-domain-containing protein	*qSNK4-2* [12]
	S05_1487229	Os05g03510	Expressed protein	
	S07_28409912	Os07g47500	Histone-arginine methyltransferase CARM1	*qRGR7* [25]
RSR	S01_23656773	Os01g41790	Expressed protein	
	S01_37680628	Os01g64910	Anthocyanidin 5,3-O-glucosyltransferase	
	S05_24090514	Os05g41130	OsFBX168-F-box-domain-containing protein	*qRRN5* [25]
	S10_18098744	Os10g35570	Expressed protein	*qSKC10.18* [2,7]
	S12_3544726	Os12g07210	Expressed protein	
log_DW	S01_36150523	Os01g62450	Expressed protein	
	S05_7195992	Os05g12510	Expressed protein	*qDLRa5-3* [24]
inv_FW	S01_3236648	Os01g06820	hcr2-0B, putative	
SKC	S09_19322095	Os09g32350	Expressed protein	*qSNC9.19* [2]
	S09_19683788	Os09g32972	MYB protein	*qSNC9.19* [2]
	S10_18834021	Os10g35230	Rf1, mitochondrial precursor	*qSKC10.18* [2,7]
SNC	S02_3477202	Os02g06890	OTU-like cysteine protease family protein	
	S03_14554651	Os03g25480	Cytochrome P450	Os03g25480 [16]
	S06_15335573	Os06g39580	Hypothetical protein	*qARL6* [38]
	S07_29627590	Os07g49470	Protein kinase APK1B, chloroplast precursor	
	S08_15439243	Os08g25390	Bifunctional homoserine dehydrogenase	Os08g25390 [16]
	S09_22076185	Os09g38340	ZOS9-17-C2H2 zinc finger protein	
SNK	S04_34643455	Os04g58160	Fiber protein Fb34, putative	*qSNK4-2* [12]

^a^ AKT, alkalinity tolerance score; SHL, shoot length; RTL, root length; RSR, root-to-shoot ratio; inv_FW, inverse fresh weight; log_DW, log dry weight; SNC, shoot Na^+^ concentration; SKC, shoot K^+^ concentration; SNK, shoot Na^+^:K^+^ ratio.

**Table 7 plants-12-02206-t007:** The mapped QTLs associated with alkalinity tolerance at the seedling stage in this genome-wide association study.

Trait ^a^	QTLs	Lead SNP	Position	*p*-Value	R^2^ (%)	QTLs in a Previous Study
SHL	*qSHL12*	S12_23108164	23,108,164	0.00058	11	-
log_DW	*qlog_DW1*	S01_36150523	36,150,523	0.00008	14	*qSHL1.38* [2,7]
SNC	*qSNC7*	S07_29627590	29,627,590	0.00027	11	-
SKC	*qSKC9*	S09_19322095	19,322,095	0.00037	22	*qSNC9.19* [2]
SKC	*qSKC10*	S10_18834021	18,834,021	0.00021	18	*qSKC10.18* [2,7]
SNK	*qSNK4*	S04_34643455	34,643,455	0.00002	16	*qSNK4-2* [12]

^a^ SHL, shoot length; log_DW, log dry weight; SNC, shoot Na^+^ concentration; SKC, shoot K^+^ concentration; SNK, shoot Na^+^:K^+^ ratio.

## Data Availability

The data presented in this study are available in the article and Appendix A.

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
