# Peer review of "Genome-Wide Association Study Identified Candidate Genes for Alkalinity Tolerance in Rice"

_plants, 2023, doi:10.3390/plants12112206_

Round 1

Reviewer 1 Report

Manuscript "Genome-wide association study identified candidate genes for alkalinity tolerance in rice" is very interesting.

General comments:
Authors evaluated a diverse panel of 184 indica and japonica rice genotypes for alkalinity tolerance at the seedling stage to evaluate the genetic variation and population structure.
Authors investigated the genetic variability, and population structure of the diversity panel and identified QTLs and candidate genes for traits associated with seedling stage alkalinity tolerance.

Detailed comments:
The 185 rice genotypes make a good population for association mapping.
Pearson's correlation coefficients were calculated for original data, or on genotypic averages?
Quality of Figure 4 is very poor. The Figure needs improvement.
Quality of Figure 5 is very poor. The Figure needs improvement.
Quality of Figure 6 is very poor. The Figure needs improvement.
Quality of Figure 8 is very poor. The Figure needs improvement.
Figure 8: "-logP' not "-log10 (P)"
Correlated traits are usually determined by the same genes. How do the authors explain the lack of such common significant markers in their study?

My suggestion:
Line 147: "[82] and" not "[82]and".
Authors used principal component analysis. This is not a good tool for data observed in repetition. The authors should use canonical variable analysis and should calculate Mahalanobis distances.
Line 518: "-logP' not "-log10 (P)"

Paper needs major revision.

Author Response

General comments:
Authors evaluated a diverse panel of 184 indica and japonica rice genotypes for alkalinity tolerance at the seedling stage to evaluate the genetic variation and population structure.
Authors investigated the genetic variability, and population structure of the diversity panel and identified QTLs and candidate genes for traits associated with seedling stage alkalinity tolerance.

The 185 rice genotypes make a good population for association mapping.

Detailed comments:

  1. Pearson's correlation coefficients were calculated for original data, or on genotypic averages?

Author’s response: The Pearson’s correlation coefficients were calculated on genotypic averages.

  1. Quality of Figure 4 is very poor. The Figure needs improvement.

Author’s response: Improved picture included as suggested by the reviewer.

  1. Quality of Figure 5 is very poor. The Figure needs improvement.

Author’s response: Improved picture included as suggested by the reviewer.

  1. Quality of Figure 6 is very poor. The Figure needs improvement.

Author’s response: Improved picture included as suggested by the reviewer.

  1. Quality of Figure 8 is very poor. The Figure needs improvement.

Author’s response: Improved picture included as suggested by the reviewer.

  1. Figure 8: "-logP' not "-log10 (P)"

Author’s response: We used built-in FarmCPU library in R software for GWAS analysis, which internally calculate the p-value using −log10 (P). Please check out the reference paper attached for more details. (https://acsess.onlinelibrary.wiley.com/doi/full/10.2135/cropsci2017.03.0160)

  1. Correlated traits are usually determined by the same genes. How do the authors explain the lack of such common significant markers in their study?

Author’s response: We tested the correlation based on phenotypic traits, not based on genotypic data. In our previous studies, we identified different QTLs for correlated phenotypic traits and reason could be different genes improving morphological and physiological traits but not correlated with each other.

Suggestions:
1. Line 147: "[82] and" not "[82]and".

Author’s response: Corrected as suggested by the reviewer.

  1. Authors used principal component analysis. This is not a good tool for data observed in repetition. The authors should use canonical variable analysis and should calculate Mahalanobis distances.

Author’s response: Our purpose in this study was to check the contribution of phenotypic traits to the variation. Most of the article use PCA for this purpose. Please see below few papers on this. Also, our aim was to visualize the relationship using 2D plots. Canonical analysis is a multivariate technique which is concerned with determining the relationships between groups of variables in a data set. It is quite difficult to offer a visual description of Canonical correlation analysis (CCA) in comparison to Principal components analysis (PCA).

Yano K. et al. (2019) GWAS with principal component analysis identifies a gene comprehensively controlling rice architecture. 116 (42): 21262-21267 https://doi.org/10.1073/pnas.190496411

Mazid MS et al. (2013) Genetic variation, heritability, divergence and biomass accumulation of rice genotypes resistant to bacterial blight revealed by quantitative traits and ISSR markers. Physiol. Plant. 149:432-447.

  1. Line 518: "-logP' not "-log10 (P)"

 Author’s response: We used built-in FarmCPU library in R software for GWAS analysis, which internally calculate the p-value using −log10 (P). Please check out the reference paper attached for more details. (https://acsess.onlinelibrary.wiley.com/doi/full/10.2135/cropsci2017.03.0160)

Reviewer 2 Report

Dear authors,

Thank you for the manuscript, Singh et al. Congratulations to the authors for finishing a daunting task of screening diverse panel of 184 indica and japonica rice genotypes 92 for alkalinity tolerance. The paper is well written and well presented.

In the introduction, please mention what is alkaline stress by its parameters like conductivity of soil. Just curious to know does water logging causes this stress?

In figure 9 apart from the LOCs can the gene names be provided along?

Based on the transcript profile can a pathway be depicted for the reason of its high tolerance of the genotypes under alkaline conditions?

Can the authors comment on the yield of these plants which are tolerant?

Were any photosynthetic parameters checked for the tolerant and susceptible genotypes?

If a better picture of the figures be provided it will be great. The legends are squished and some figures not readable.

Author Response

Thank you for the manuscript, Singh et al. Congratulations to the authors for finishing a daunting task of screening diverse panel of 184 indica and japonica rice genotypes 92 for alkalinity tolerance. The paper is well written and well presented.

  1. In the introduction, please mention what is alkaline stress by its parameters like conductivity of soil. Just curious to know does water logging causes this stress?

Author’s response: Updated with information on characteristics of alkali soil as suggested by the reviewer. Waterlogging normally bring toxic salts to the crop root-zone and can make soil more alkaline. However, it depends if these salts are carbonates or bicarbonates or neutral salts.

The accumulating salts also turn the soil more alkaline and hamper the growth of crops.

  1. In figure 9 apart from the LOCs can the gene names be provided along?

 Author’s response: Updated as suggested by the reviewer.

  1. Based on the transcript profile can a pathway be depicted for the reason of its high tolerance of the genotypes under alkaline conditions?

Author’s response: In our study, we did expression profiling of only eight genes by qRT-PCR. It is difficult to pinpoint the specific pathways associated with alkali tolerance. However, based on the genes harboring the SNPs significantly associated with alkalinity tolerance traits, many of these were involved in ion transport and signal transduction. To gain more understanding of the pathways involved in alkalinity tolerance, a comparative transcriptomic profiling of tolerant and susceptible genotypes will be required.

  1. Can the authors comment on the yield of these plants which are tolerant?

Author’s response: Thanks for the suggestion. We used diverse germplasm to test the tolerance under alkaline stress at the seedling stage. These plants have not been tested at the reproductive stage under alkaline stress.

  1. Were any photosynthetic parameters checked for the tolerant and susceptible genotypes?

Author’s response: No, we did not check any photosynthetic parameters in this study.

  1. If a better picture of the figures be provided it will be great. The legends are squished and some figures not readable.

Author’s response: Pictures updated as suggested by the reviewer.

Round 2

Reviewer 1 Report

The notation "-log10 (P)" and "-logP" mean the same thing. In mathematics, with decimal logarithms, the number '10' is not written. The figure should be corrected. Claiming that a program does so is not sufficient. Correct records should always be used, not those declared by the package.

Of course, correlation is tested based on phenotypic traits, not on genotypic data. Phenotypic traits are continuous, while genotypic traits are discrete. It is very common in the subject literature to report that phenotypically correlated traits are determined by the same genes. In the presented study, different results were obtained. This should be commented on and discussed with the results of other researchers.

To say that most (surely most?) articles are based on some method is not a scientific approach. One can point to a great many articles that apply canonical variable analysis. The authors mention that their goal was to test the contribution of phenotypic traits to variation. PCA is based on mean values, while CVA takes into account within-subject variability, which the authors want to show. The authors should use canonical variable analysis and calculate Mahalanobis distances!

Paper needs major revision. The authors should consult a statistician.

Author Response

Reviewer Comment 1: The notation "-log10 (P)" and "-logP" mean the same thing. In mathematics, with decimal logarithms, the number '10' is not written. The figure should be corrected. Claiming that a program does so is not sufficient. Correct records should always be used, not those declared by the package.

Authors response: Figure was revised to include -log(p) instead of -log10(p) as suggested by the reviewer..

Reviewer Comment 2: Of course, correlation is tested based on phenotypic traits, not on genotypic data. Phenotypic traits are continuous, while genotypic traits are discrete. It is very common in the subject literature to report that phenotypically correlated traits are determined by the same genes. In the presented study, different results were obtained. This should be commented on and discussed with the results of other researchers.

Authors response: Thanks for the suggestion. Normally, there are two common reasons such as pleiotropy (same gene controlling multiple traits) or tight linkage of genes when correlations are observed among the phenotypic traits. But in our case our results are in contrast the above notion i.e. traits were correlated but not controlled by the different QTLs/genes. We have discussed with few examples (Descalsota et al. 2018; Alkahtani et al. 2022) where no QTLs were detected for correlated traits. In our case, apart from genetic complexity of traits contributing toward alkalinity tolerance, low density of SNPs might have contributed to failure to detect SNPs for the correlated traits.

Reviewer Comment 3: To say that most (surely most?) articles are based on some method is not a scientific approach. One can point to a great many articles that apply canonical variable analysis. The authors mention that their goal was to test the contribution of phenotypic traits to variation. PCA is based on mean values, while CVA takes into account within-subject variability, which the authors want to show. The authors should use canonical variable analysis and calculate Mahalanobis distances! Paper needs major revision. The authors should consult a statistician.

Author’s response: Thank you for the suggestion. We appreciate your input. Our objective in conducting this analysis was not only to assess the individual contributions of traits but also to compare the differences between the indica and japonica groups. We firmly believe that the Principal Component Analysis (PCA) approach to examine trait contributions and employing scatter plots to visualize group differences (Figure 4), is appropriate for our study. Furthermore, we consulted a professional statistician who recommended the use of PCA in our case. We are confident that the purpose of conducting PCA justifies the results we have obtained. While we acknowledge that Canonical Variate Analysis (CVA) is an alternative technique, we did not find a substantial body of literature that extensively utilized CVA in studies similar to our case. Therefore, we believe that maintaining the current structure of our manuscript aligns with the prevalent literature and provides meaningful insights into genetics of alkalinity tolerance for the readers.

Round 3

Reviewer 1 Report

Authors have incorporated all the suggestions, accordingly. I recommend this article to publish in current version.